# Role of Machine Learning-Based CT Body Composition in Risk Prediction and Prognostication: Current State and Future Directions

**DOI:** 10.3390/diagnostics13050968

**Published:** 2023-03-03

**Authors:** Tarig Elhakim, Kelly Trinh, Arian Mansur, Christopher Bridge, Dania Daye

**Affiliations:** 1Department of Medicine, Perelman School of Medicine at the University of Pennsylvania, Philadelphia, PA 19104, USA; 2Department of Radiology, Massachusetts General Hospital, Boston, MA 02114, USA; 3School of Medicine, Texas Tech University Health Sciences Center, School of Medicine, Lubbock, TX 79430, USA; 4Harvard Medical School, Harvard University, Boston, MA 02115, USA

**Keywords:** artificial intelligence, machine learning, CT body composition, prognostication, risk prediction

## Abstract

CT body composition analysis has been shown to play an important role in predicting health and has the potential to improve patient outcomes if implemented clinically. Recent advances in artificial intelligence and machine learning have led to high speed and accuracy for extracting body composition metrics from CT scans. These may inform preoperative interventions and guide treatment planning. This review aims to discuss the clinical applications of CT body composition in clinical practice, as it moves towards widespread clinical implementation.

## 1. Introduction

Body mass index (BMI) has long been a key clinical metric that is used in predictive models to estimate the risk of developing chronic diseases and future mortality [1]. Unfortunately, BMI has several shortcomings and does not account for the distribution of fat in the body and does not distinguish between excess fat, proportion of bone mass or muscle [2]. Because BMI only measures excess weight, this measure cannot reflect the loss of muscle mass as in sarcopenia and sarcopenic obesity. Body composition metrics incorporate the proportion of body fat and skeletal muscles. Many non-invasive measures exist that analyze body composition starting from the traditional skin-fold tests to other advanced measures such as bioelectrical impedance analysis, dual-energy X-ray absorptiometry (DXA), hydrostatic (underwater) densitometry and air displacement plethysmography, among others [3]. Despite some studies suggesting their accuracy compared to BMI, many of these latter metrics are considered inappropriate for widespread clinical implementation because they are often complex to implement, expensive, and difficult to standardize [2]. As such, the CDC currently only recommend BMI as an indication of body composition and health risks, especially as other measures have no available reference standards or validated risk categories [2].

Advances in CT scan automated segmentation and deep learning have opened the door for the implementation of CT body composition as a novel tool to assess health and disease risk (Figure 1) [4]. Multiple studies have shown the clinical importance of CT body composition in risk prognostication and treatment planning [5]. We project that these new metrics can eventually replace BMI in many clinical applications. Many patients currently undergo a CT scan for diagnostic purposes. Much data from these scans are not being used for clinical decision-making. CT scans have the capability of providing more information, in addition to their specific clinical indication, allowing opportunistic screening for disease prognostication and primary prevention. Recently, there have been new approaches that have decreased radiation exposure per unit volume of imaging, making CT more suitable [6,7,8]. Many targeted interventions can be applied to improve disease prevention and eventually patient outcomes.

## 2. Imaging-Based Body Composition Analysis

CT and MRI are regarded as the gold standard for body composition analysis [9,10,11] and can be used to quantify body composition. While these methods are costly, multiple people undergo cross-sectional imaging for other clinical indications, allowing for opportunistic assessment of body composition. CT works by taking multiple X-rays of the body from various angles while MRI uses the magnetic properties of hydrogen nuclei in the cells of the body to create images of soft tissues. Both methods allow for detailed evaluation of individual skeletal muscles and adipose tissue, although one study showed that MRI slightly underestimates visceral adiposity [12]. There have been several attempts to compare the performance of CT and MRI in CT body composition analysis, with most studies showing a high correlation between both modalities [13,14]. However, MRI voxel values are highly dependent on a number of factors related to the interactions between protons [12].The inconsistency in voxel values is one of the biggest challenges for MRI, thus making advancement of this approach more difficult. Despite MRI not exposing patients to ionizing radiation, the use of CT is considered quick, easy and less costly. Additionally, CT imaging is widely available compared to MRI. In a recent 2000–2016 analysis of seven US healthcare systems, CT annual imaging rates in the US have increased from 56 to 141 per 1000 person per year, while MRI increased from 16 to 64 per 1000 person per year [15]. This shows the widespread availability of CT imaging, making it a more powerful tool for opportunistic screening, population-based analysis and large scale investigations. As a result, multiple AI-based algorithms have been developed to perform automated CT body segmentation and quantification by measuring skeletal muscle and fat, typically at the L3 vertebrae. These values are then used to estimate whole-body composition and prognosticate patients.

Below, we summarize the standard metrics extracted from CT body composition algorithms and expand on their clinical applications (Table 1).

### 2.1. Muscle Mass

Low skeletal muscle is termed sarcopenia and has been associated with worse clinical outcomes in conditions such as cancer [16], cirrhosis [17] and critical illness [18], among others as well as postoperatively [19,20]. Sarcopenia has primarily been found to be an independent predictor of survival in cancer patients [44]. In one study, cancer patients who did not look thin or malnourished were found to have sarcopenia only through CT body composition analysis [45]. In a recent systematic review, eight studies showed that the reduced muscle mass was mainly detected through CT body composition analysis with a high number of patients being misclassified based on BMI [46]. CT body composition detects sarcopenia at a rate that is 27.3–66.7% higher compared to the detection of malnourishment using BMI.

Sarcopenia can be difficult to assess clinically even with the use of BMI. For instance, some patients have a high proportion of fat to muscle ratio as seen in sarcopenic obesity. In another study, obese patients with a BMI > 30 mg/m were found to be sarcopenic through CT body composition [47]. Sarcopenic obesity is the extreme of two phenotypes being low muscle mass and high BMI. It has been associated with worse clinical outcomes, especially in cancer patients [48]. This type of abnormal body composition is not detected clinically because muscle and fat tissue quantification is required to establish the diagnosis. Multiple studies have showed that sarcopenia can occur through all ranges of patients’ BMI [46]. CT body composition can better identify patients at risk of worse clinical outcomes [49]. To diagnose sarcopenia, the European Consensus Statement now recommends using a CT scan as the gold-standard technique [50], highlighting the importance of CT body composition and its potential in clinical practice.

There are several ways suggested to diagnose sarcopenia on CT imaging. In a systematic review and meta-analysis of 70 studies from 15 countries that used CT to assess sarcopenia, 88.4% used skeletal muscle index (SMI) L3 to diagnose sarcopenia, five used visceral fat criteria and three used the total psoas area (TPA) criteria [51]. SMI is determined by measuring the total skeletal muscle area (cm^2^) at the L3 level and dividing by the height squared (m^2^). Among the studies that used SMI, there were several cutoff criteria used. The three most common include: (1) the cut-offs introduced by Prado et al., which defined sarcopenia as SMI < 52.4 cm^2^/m^2^ for males and < 38.5 cm^2^/m^2^ for females, which have been used in 20 studies [49]; (2) the cut-offs introduced by Martin et al., which defined sarcopenia as SMI < 53 cm^2^/m^2^ if BMI ≥ 25 kg/m^2^ or SMI < 43 cm^2^/m^2^ if BMI < 25 kg/m^2^ in males and SMI < 41 cm^2^/m^2^ in females, and have been used in 17 studies [45]; (3) those introduced by Zhuang et al., which defined sarcopenia as SMI < 40.8 cm^2^/m^2^ in males and SMI < 34.9 cm^2^/m^2^ in females, and have been used in 12 studies [52]. Of the studies that used the visceral fat criteria, most used the cut-off from the Japanese Society for the Study of Obesity which describes a visceral fat area (VFA) of ≥100 cm^2^ as the cutoff [53]. Of the studies that used the TPA criteria, most used the cut-off of Fearon et al. which defined sarcopenia by calculating the total cross-sectional area (mm^2^) of the psoas muscle at L3 and dividing by height squared (m^2^). Its cutoffs have an international consensus defined as <385 mm^2^/m^2^ in women and <545 mm^2^/m^2^ in men [54].

### 2.2. Skeletal Muscle Quality

SMI and skeletal muscle radiation attenuation (SM-RA) obtained from CT scan allows the evaluation of myosteatosis or low muscle quality. Deposition of fat in muscles is indicative of muscle wasting. A recent study of HCC patients undergoing hepatectomy found that myosteatosis is associated with worse perioperative morbidity, mortality and long-term oncological outcomes compared to sarcopenia [55]. Myosteatosis was also found to have an important prognostic role in HCC patients undergoing surgery and can be an independent risk factor of perioperative morbidity. Assessment of myosteatosis is important to complement other body composition metrics to predict perioperative and long-term disease outcomes.

### 2.3. Visceral Fat Content

High visceral fat is associated with increased systematic vascular resistance, lower cardiac output, insulin resistance and higher pro-inflammatory factors promoting carcinogenesis [56,57]. The most commonly implicated inflammatory markers are tumor necrosis factors, interleukin-6, adiponectin and free fatty acids [58] which are found to directly flow through the portal vein causing liver inflammation, NASH cirrhosis and hepatocellular carcinoma [35]. High visceral fat was also found to be an independent predictor of major cardiovascular events, cancer risk, metabolic syndrome and mortality in asymptomatic screening populations and in patients with colon cancer [36,37,38,59]. This emphasizes the importance of analyzing visceral fat mass, in addition to skeletal muscle, to aid in predicting overall health and outcomes from various diseases and therapies [45,47].

### 2.4. Bone Density

CT body composition examinations typically incorporate information on bone density, providing CT-based opportunistic screening for osteoporosis [60]. Due to its volumetric nature, CT images may be more accurate in determining bone mineral density compared to DEXA [61]. Thus, the development of an algorithm that is capable of segmenting CT images automatically and accurately can assist in predicting future risk of osteoporotic fractures. Pickhardt et al. used an automated, feature-based image processing algorithm to measure L1 trabecular attenuation, and the result was consistent with data from manual region-of-interest placement [62]. Tan et al. created an automated algorithm to segment vertebral body for measurement of syndesmophytes and progression of ankylosing spondylitis [63]. Opportunistic screening for reduced bone density can be performed simultaneously as patients undergo CT scanning for other indications [64]. As such, CT body composition may allow for the opportunistic detection of osteoporosis and may potentially improve access to early treatment and management.

### 2.5. Arterial Calcifications

Coronary artery calcification can be quantified using CT body composition software for opportunistic screening. Studies have shown a strong correlation between coronary artery calcification score and future cardiac events [65,66]. Additionally, abdominal CT scan can quantify abdominal aorta calcifications, which are found to have positive correlation with coronary heart disease [39,67]. Pickhardt et al. developed a deep-learning mask region-based convolutional neural networks (R-CNN) algorithm to segment and quantify calcified atherosclerotic plague within the abdominal aorta from CT scans [68]. The algorithm automatically selects the L1–L4 vertebral levels to perform segmentation and quantification of aortic calcification. CT-based abdominal aorta calcification scores obtained from both semi-automated and automated methods have been shown to better predict future cardiovascular events compared to the Framingham Risk Score [40].

### 2.6. Other CT-Based Quantitative Metrics

Several other studies have recently shown that many additional quantitative parameters can be extracted from CT images. CT allows the quantification of epicardial adipose tissue, which is the biologically active adipose tissue between the myocardium and the visceral pericardium that is associated with adverse cardiovascular events [10,41]. Given the unreliability of creatinine excretion and eGFR equations for patients with certain body compositions, Pieters et al. developed equations that estimated creatinine production by using deep-learning body composition analysis of CT images [42]. Abdominal CT biomarkers, such as pancreatic CT attenuation, fat content and fractal dimension, can also be assessed with deep learning, and in particular can aid in the diagnosis of type 2 diabetes mellitus [43]. Additionally, CT scans allow for the segmentation of organs at risk, which is imperative for planning radiotherapy [69]. Deep learning methods have been developed to automate organ segmentation, such as in the parotid gland [70], prostate [71], adrenal gland [72], mammary glands [68] and multiple other multi-organs [73,74]. Other emerging targets for CT-based analysis using automated segmentation include the detection and assessment of intracranial internal carotid artery calcification [75]. Cui et al. developed an automated segmentation algorithm using dense V-networks for small gross tumor volumes in lung cancer from 3D planning CT images [76]. Lin et al. proposed a 3D UNet-based deep learning model for automated segmentation and detection of renal tumors [70]. Their newly created model has shown promising results with high levels of accuracy. Bilic et al. reviewed and analyzed around 75 state of the art automated liver and liver tumor segmentation algorithms from CT scans and found that the best liver segmentation algorithm achieved a dice score of 0.963, but for liver tumor segmentation the highest achieved was a dice score of 0.739, indicating further research need in this area [77]. With new advances in deep learning and image segmentation, we envision that new metrics will be made available over the coming years to be used in clinical practice to improve patient disease screening and management.

## 3. Clinical Applications of CT Body Composition

### 3.1. Cancer

Sarcopenia is associated with increased morbidity and mortality in multiple types of cancer [78] including pancreatic cancer [79], esophageal cancer [80], lung cancer [81], colorectal liver metastasis [82] and melanoma [83], among others. In a recent systematic review of CT body composition in abdominal malignancy, seven studies showed that low muscle mass was associated with a worse clinical outcome [46]. Sarcopenia was linked to adverse therapeutic and clinical outcomes including higher postoperative infections, systematic inflammation, chemotherapy toxicity and mortality in patients with abdominal malignancy [46].

In another multi-center retrospective study of preoperative CT body composition analysis in lung cancer patients undergoing lobectomy, skeletal muscle mass was an independent predictor of postoperative complications and increased hospital length of stay (LOS) [84]. Interestingly, low thoracic muscle mass was more effective than biological age in predicting postoperative events [85]. In the same population, sarcopenic obesity was an independent predictor of hospital LOS and postoperative complications. This highlights the role of CT body composition in identifying cancer patients who carry a high risk of worse clinical outcomes prior to surgery.

Similar results have been reported in patients with hepatocellular carcinoma (HCC) [58]. CT body composition has been found to be predictive of patient outcomes in those receiving chemotherapy, radiotherapy, radio-frequency ablation, embolization, hepatectomy and liver transplant [44]. In a recent study evaluating the prognostic factors associated with overall survival in elderly patients with HCC receiving trans-arterial chemoembolization (TACE), the detection of muscle depletion and visceral adiposity was found to be independently associated with poor survival outcomes [86]. The same study found no relationship between BMI and survival [87]. Interestingly, the response to the first TACE session was better in those with low muscle mass and high visceral fat compared to those with normal body composition [87]. However, the former group had lower overall survival. As such, assessment of body composition may be an important clinical consideration for HCC patients undergoing TACE. Similarly, Faron et al. evaluated the role of sarcopenia to predict overall survival in those receiving yttrium-90 (Y90) trans-arterial radioembolization (TARE) [88]. Sarcopenia was found to be an independent prognostic marker of overall survival and can provide prognostic value in patients receiving Y90 TARE [89]. Another study assessed sarcopenia before and after treatment with TARE and found it to be predictive of post-TARE progressive HCC disease [90]. Similarly, HCC patients with sarcopenia undergoing radiofrequency ablation therapy were found to have a lower survival rate compared to nonsarcopenic patients [91].

In HCC patients undergoing hepatectomy, sarcopenia was associated with high rates of post-surgical complications [92,93]. One study showed that the 5-year survival rate was lower in those with sarcopenia compared to non-sarcopenic patients (58.2% vs. 82.4%, *p* = 0.0002) [94]. Additionally, having sarcopenia was associated with a worse tumor stage and microvascular invasion [95]. Another study showed that patients with sarcopenia have higher rates of morbidity and mortality after hepatectomy [85], similar to those who have diminished functional reserves [87]. When considering hepatectomy, it is important to assess the future liver remnant (FLR), the volume of liver to be left behind after resection [89]. Those with small FLR have a higher risk of post-hepatectomy liver failure [96]. Many of these patients undergo portal vein embolization (PVE) prior to hepatectomy so as to divert portal venous blood and trophic factors to the non-embolized section of the liver leading to liver hypertrophy of the non-resected liver segments. Those with insufficient hypertrophy are at increased risk of post-hepatectomy liver failure [97,98]. A recent study evaluated the role of CT body composition in predicting liver remnant hypertrophy following PVE in patients with colorectal liver metastasis. The study found that patients with sarcopenia had impaired liver hypertrophy after PVE [99]. Another study also found that the quantity and quality of skeletal muscle were associated with the degree of liver hypertrophy after PVE [95]. Low muscle mass on CT body composition was found to be an independent predictor of poor liver hypertrophy after PVE and increased the risk of post-hepatectomy liver failure [100]. These studies suggest that the assessment of CT body composition prior to PVE may be important for identifying patients at risk of post-hepatectomy complications.

In addition to the prognostic association of sarcopenia with poor performance status, cancer progression and overall survival, it has also been linked to chemotherapy toxicity and response to therapy [101,102]. A recent retrospective study found decreased survival rates in sarcopenic patients receiving sorafenib chemotherapy for HCC compared to nonsarcopenic patients [103]. Additionally, sarcopenic patients were found to have a lower response to chemotherapy and lower disease control compared to nonsarcopenic patients [104]. Another study found sarcopenia to be associated with early dose chemotherapy toxicity [105]. These results raise the question of possible future adjustments of chemotherapy dose based on the amount of skeletal muscles that a patient has, to avoid extensive toxicity [106].

### 3.2. Liver Disease

Studies have shown an association between CT body composition and severity of liver disease [100,107,108]. Liver cirrhosis is strongly associated with sarcopenia [109]. The distribution of body fat is a major predictor of complications and outcomes in patients with cirrhosis, both before and after liver transplantation [110]. Therapy for liver disease is also associated with alterations in body composition. For instance, transjugular intrahepatic portosystemic stent (TIPS), a standard therapy in many patients with portal hypertension, is associated with improved fat-free mass and fluid-free body weight [104,111,112]. Artu et al. utilized CT scans to measure body composition in patients post-TIPS placement and found an improvement in sarcopenia and decreased visceral-to-subcutaneous fat ratio following intervention [113]. Additionally, Pang et al. were able to demonstrate that pre-TIPS blood ammonia had a positive association with post-TIPS BMI [114]. These studies demonstrate the importance of CT body composition analysis before and after treatments in liver disease.

In addition to its association with response to therapies, CT body composition can also be used to predict the etiology of liver disease. Zou et al. developed a deep learning algorithm using Google’s DeepLabv3+ in which body composition was automatically extracted [115]. Their study showed that patients with NAFLD cirrhosis had decreased muscle mass and a significant increase in visceral and subcutaneous fat compared to those with non-NAFLD cirrhosis. The study also showed higher levels of accuracy of CT body composition compared to that of BMI in distinguishing the two patient populations. These findings highlight the potential role of CT body composition in risk prediction and stratification in liver disease.

### 3.3. Inflammatory Bowel Disease (IBD)

Analysis of abdominal CT body composition can also aid in disease prognostication in patients with Crohn’s Disease and ulcerative colitis (IBD). IBD is a gastrointestinal inflammatory disorder associated with malabsorption resulting in low skeletal muscle mass, decreased bone mineral density and therefore a dynamic change in body composition especially in patients with Crohn’s disease [116]. Abdominal CT-based opportunistic screening has been utilized in several studies for prognostication in IBD. Changes in CT body composition metrics in patients with IBD are correlated with disease duration and severity [117]. The pathogenesis of Crohn’s disease is associated with increased visceral adiposity as identified through CT body composition. In patients with increased visceral adiposity, studies have reported a more complicated disease course [118], higher postoperative complication rates [119] and higher rates of disease recurrence [120], Grillot J et al. also reported worse Crohn’s disease outcomes with sarcopenia and visceral adiposity [114]. Another similar study found that muscle volume is strongly associated with hospital length of stay and that both, muscle volume and visceral adiposity, are strongly associated with intestinal resection rates [121]. These results highlight that early screening and detection of body composition changes in patients with Crohn’s disease may help in risk stratification and may inform early nutritional and pharmacological interventions, potentially improving patients’ outcomes and quality of life [122].

### 3.4. Kidney Disease

Paradoxically, higher BMIs are associated with better survival in patients with chronic kidney disease (CKD) [123,124,125]. However, due to the limitations of BMI, it is not fully known whether the increase in survival is associated with levels of adipose tissue or lean mass. Patients with CKD tend to have fluid retention that cannot be differentiated with BMI. Lin et al. showed, through using a body composition monitor–multifrequency bioimpedance spectroscopy device, that a high lean tissue index, not high BMI or high fat tissue index, predicted a lower risk of adverse outcomes in CKD patients [126]. These findings illustrate the importance of body composition analysis and its association with outcomes in patients with kidney disease.

Fully automated CT-based body composition analysis shows great promise as it can detect total muscle mass and quantify muscle wasting which is frequently seen in this patient population [127]. It has been already shown that body composition analysis can accurately predict urinary creatinine excretion, creatinine clearance, and glomerular filtration rate (GFR) [21]. A recent study showed that machine-learning CT body composition analysis can estimate creatinine excretion with a high degree of accuracy [75]. These fully automated body composition analyses can validate Chronic Kidney Disease Epidemiology Collaboration (CKD-EPI) equation results and replace burdensome 24-h urine collection with spot urine collection, paving the way for integrated diagnostics that use multidisciplinary data for better patient care [128,129]. Furthermore, there have been recent efforts to fully automate kidney segmentation by measuring kidney, cortex and medulla volumes, which will provide a wide range of clinical applications such as evaluating renal donor suitability and prognosticating outcomes [130].

Other studies have found a correlation between high visceral adipose tissue and poor outcomes in patients with kidney disease [131,132]. Sarcopenia was also found to have a strong association with increased mortality and morbidity in patients with this condition [133]. Other studies have shown that skeletal muscle and visceral adipose tissue derived from CT scans are stronger predictors of renal disease prognosis and can outperform established clinical parameters for risk stratification [134]. In summary, utilization of CT body composition to accurately quantify muscle mass and calculate visceral-to-subcutaneous fat ratio has the capability of aiding prognostication in patients with renal disease.

### 3.5. COVID-19

Several studies have shown the association between CT body composition parameters and the severity of COVID-19 disease. Hocaoglu et al. and Ufuk et al. utilized CT to measure pectoralis muscle volume and density. They found that low pectoralis muscle density correlated with increased COVID-19 severity and worse outcomes [135,136]. Chandarana et al. showed that CT-derived muscle adipose tissue measurements at the L3 vertebral level were significantly higher in patients with more severe symptoms of COVID-19; consequently, those patients had a higher risk of hospitalization [137]. Similarly, Bunnell et al. performed body composition segmentation using an in-house automated algorithm trained specifically at the L4 vertebral level and found that COVID-19 patients with high visceral adipose tissue/subcutaneous adipose tissue ratio and high intermuscular adipose tissue have worse outcomes [138]. Another study analyzed paravertebral muscle at the 12th thoracic vertebra in COVID-19 patients and found that muscle loss is a predictor of intensive care admission in COVID-19 patients. Taken together, these findings suggest that CT body composition analysis can help predict adverse clinical events and outcomes in patients with COVID-19.

### 3.6. Cardiovascular Diseases

Cardiovascular disease (CVD) remains the leading cause of morbidity and mortality worldwide [139]. CT-based opportunistic screening can help detect cardiovascular diseases pre-symptomatically, thus allowing early preventative care to decrease future adverse clinical events and healthcare costs. O’Connor et al. showed that the abdominal aortic calcification score using semiautomated CT quantifications is a better predictor of cardiovascular events than the Framingham risk score (FRS) [73]. Other studies have shown that controlling the progression of abdominal aortic calcification was associated with decreased risk of mortality, coronary artery disease, stroke and heart failure [140,141]. By detecting aortic calcification early using CT-based opportunistic screening, appropriate interventions can be applied to those patients to address their underlying risk and prevent future cardiovascular mortality. Similarly, Pickhardt et al. defined several automated CT-based body composition biomarkers that can predict major cardiovascular events, including quantification of aortic calcification, muscle density, visceral/subcutaneous and liver fat and bone mineral density. These metrics outperformed clinical parameters such as the FRS and BMI for risk prediction [65]. Recently, Magudia et al. described a retrospective study of 9752 outpatient routine CT scans of black people and white people with no recent history of cancer or cardiovascular diseases [142]. Using a fully automated AI approach, the SMA, VFA and SFA were extracted from the L3 vertebra, then adjusted to age, race and sex, and associated with subsequent myocardial infarction and the risk of stroke within 5 years from the scan. Interestingly, the VFA had a significant association with the risk of developing MI (HR 1.31, *p* = 0.04) and Stroke (HR 1.46, *p* = 0.04) while BMI, weight, SFA and SMA had no association. This suggests the importance of incorporating SFA instead of BMI in cardiovascular risk models.

By providing a better assessment of a person’s cardiometabolic profile, CT-based body composition analysis shows great promise than established clinical parameters in improving pre-symptomatic detection and risk-stratification of patients vulnerable to adverse cardiovascular events and can augment the current risk prediction models.

### 3.7. Critical Illness

CT body composition also plays a role in improving care in critically ill patients. Toledo et al. demonstrated that critically ill patients with sarcopenia have a lower 30-day survival, higher hospital mortality, and higher complication rates [22]. Weijs et al. reported that sarcopenia on CT, during early stages of a critical illness, is strongly associated with a high risk of mortality in mechanically ventilated critically ill patients [23]. Early identification of at-risk patients can help inform any necessary interventions for better outcomes in this critically ill population.

### 3.8. Contrast Dose Adjustment

Iodinated contrast dosing is currently calculated based on total body weight, regardless of adipose and muscle content. However, patients with various body composition indexes, such sarcopenic obesity and athletes with high muscle content, can suffer from overdosing or underdosing. To alleviate this concern, CT body composition analysis has been shown to allow appropriate contrast dosing for each patient during the process of CT scanning [28].

## 4. CT Body Composition Analysis—Technical Considerations

In this section, we discuss a number of technical considerations associated with the creation of computational methods for body composition analysis from CT. The first step in CT body composition analysis is identification of the most appropriate location for extracting body composition parameters, followed by the segmentation of the structures of interest. Although most use a single axial CT image slice, commonly located at the level of the L3 vertebra, some discussed the benefit of 3D-based analysis of CT body composition. Selecting a single slice for analysis simplifies the process of machine learning model development both by reducing the annotation burden required for training and validating the model, and reducing the complexity of the segmentation model itself. Furthermore, using a single slice eliminates potential variability in volumetry due to different spatial extents of scans, which may otherwise complicate analysis. However, single slice analysis is an imperfect proxy for overall body composition and small shifts in the location and angle of the chosen slice introduce significant variability in the measurements, particularly when slice selection is an automated process with its own error rate [29,143]. By contrast, 3D analysis provides a more comprehensive characterization of body composition. Koitka et al. described a fully automated U-Net 3D neural network for volumetric body composition analysis using every fifth axial slice from abdominal CT scans to include multiple body regions [144]. The network produced an excellent result with a Sørensen Dice coefficient of 0.9553 for segmentation and an intra-class correlation coefficient of 0.99 on tissue volumetry.

Adipose tissue is especially sensitive to the angle cut of the slice image; therefore, its measurements can change dramatically from one slice to another. Although this is insignificant with population studies, it remains a bias for individual predictions and decision making especially when analyzing individuals over time [145]. Shen et al. showed that in group studies, the use of an appropriate single slice analysis compared to volumetric body composition analysis requires 17% and 6% more subjects for estimating whole body muscle and fat, respectively [146]. As such, with large scale investigations the use of a single slice analysis is appropriate since it decreases the cost and complexity of the analysis whereas the power of the study can easily be increased with a higher number of subjects. That said, for individual decision making and small-scale investigations, the use of 3D-based analysis remains a better choice as it provides a topographic information of various tissues and accurately distinguishes individual variability.

The wealth of information obtained from 3D-based analysis introduced the concept of “extended body composition” that entails the measurement of multiple organs in the body and can identify the exact individual phenotype for better decision making and treatment selection [145]. Because a single slice ca not provide the same detailed measurements of various tissues in the body, the choice of slice location becomes crucial for accurate estimation of whole body composition and predicting patient outcomes. Shen et al. showed that CT scans around the L3 lumbar vertebra have a strong association with the composition of subcutaneous fat tissue, visceral tissue and skeletal muscles in the body [147]. Similarly, other studies also showed the accuracy of CT body composition analysis at the level of the third lumbar vertebra (L3) and have established a fully automated deep-learning system for L3 selection and body composition analysis [148]. Others found an excellent correlation between T12 and L3 for estimating body composition and argued against the need for abdominal CT imaging specially when chest imaging is the only option available [149]. Another recent study showed that the aggregation of skeletal muscle from different vertebral levels can better prognosticate and predict patient outcomes [150].

Following the choice of location for CT body composition analysis comes the role of segmentation for biomarkers’ extraction and analysis. Segmentation is executed either manually or by using automated segmentation techniques. For manual segmentation, trained image analysts or board-certified radiologists determine the region-of-interest, then select slices and distinguish each body compartment (muscle, visceral fat, subcutaneous fat) using anatomic knowledge and tissue-specific Hounsfield Unit ranges, then each slice is manually segmented [145,151,152]. Since analysts must review and segment each selected slice, the process of manual CT body composition analysis becomes challenging in a large dataset as it requires time and expertise. This limits large scale investigations from being easily performed to expand its clinical value. To overcome this issue, the new mainstay technique for CT body composition analysis uses automated segmentation. Automated and accurate CT-scan segmentation of subcutaneous fat tissue, visceral fat tissue and skeletal muscle through artificial intelligence has been reported by multiple studies [147,153,154]. Segmentation of multiple tissues can be obtained accurately using the same neural network [153,155] which provides a faster computation speed of analysis with great accuracy. Studies have reported that the analysis of CT body composition takes around 15 min/scan for a human analysis, vs. <1 s/scan with the use of neural networks [4]. This higher speed of analysis has made automated machine learning-based analysis the preferred method for large-scale investigations.

Multiple studies have established automated machine learning algorithms for CT body composition analysis [156]. Convolutional neural networks, and in particular the U-Net architecture, a well-established convolutional neural network (CNN) architecture for various medical image segmentation tasks, are the foundation of most methods. Notable examples are summarized below.

Paris et al. established a new convolutional neural network (CNN), AutoMATiCA, for the segmentation of body composition that quantifies Skeletal Muscle (SM), intermuscular adipose tissue (IMAT), Visceral Adipose Tissue (VAT) and Subcutaneous Adipose Tissue (SAT) at the L3 vertebral body. The algorithm is a combination of four separate neural networks representing four different body compartments. Their results suggest that the algorithm may be generalizable to other populations for body composition calculation [157]. Similarly, Hsu et al. developed a CNN model based on the U-Net architecture to quantify VAT, SAT and SM at the L3 level, with results consistent with the results obtained through manual segmentation [158]. CNNs were also adapted for the development of automated segmentation in the work of Weston et al. The algorithm performs as well as expert manual segmentation [155]. Bridge et al. developed a fully machine-operated algorithm to segment body composition from an abdominal CT scan [153]. The method is broken down into two steps: (1) automatically identify and select a slice at L3 vertebral level from a full CT scan; and (2) segment body composition using a U-Net-based segmentation network. Dice score results were comparable between the AI-based segmentation and manual segmentation [153]. They later extended the same approach to three thoracic levels (T5, T8, and T10) [159]. Many other works have similarly demonstrated successful segmentation of body composition using automated approaches [154,155,156,160,161,162,163].

With further advancement in this field, it became evident that there is a need to establish reference ranges and adjust for body composition values based on demographic variables such as age and gender. Recent studies performed population-scale CT body composition analysis and established age-, sex-, and race-specific reference curves for CT body composition metrics, analogous to reference ranges for the Z-scores used in DEXA scans [164]. CT body composition reference parameters were found to be different across demographic groups, unlike the traditional reference ranges for BMI and weight metrics. In the same work, the derived CT body-composition Z-scores were found to be predictive of patient survival, further strengthening the value of CT body composition analysis in clinical care (Figure 1).

## 5. Future Directions

CT imaging provides physicians with many datapoints, beyond the scan’s clinical indications. The plethora of data available from CTs have sometimes been viewed unfavorably due to the concern of incidental findings triggering unnecessary workups. However, there has also been a concordant rise in interest of CT-based opportunistic screening due to its ability to identify at-risk patients and avoid future adverse events [39]. While CT imaging allow for a whole range of body composition analyses, such as quantifying bone mineral density for osteoporosis or analyzing visceral fat for metabolic syndrome, the additional data are often not utilized in routine clinical care [39]. This under-utilization was likely due to the labor-intensiveness of manual or semi-automated body composition analyses, especially when outside the clinical indication of the scan. With the recent innovations in fully automated methods of CT body composition analyses, this technique is now more readily accessible.

Beyond the traditional CT body composition metrics presented in this article, the rise in fully automated AI-based methods for analyzing CT scans has further expanded our capabilities through additional organ-specific segmentation and detection [165]. For instance, new methods are now available to determine abdominal organ volume for organomegaly [166], to stage liver fibrosis [167] and to detect tumors [168], among others [169]. These new methods present a paradigm shift in how clinicians will be able to use cross-sectional imaging for clinical management, revolutionizing the state-of-the-art care that patients can receive.

Despite the promising potential that fully automated AI-based CT body composition analysis brings to the field, its application remains dependent on robust data analysis and large-scale investigations to validate its clinical importance and strengthen its value in the clinical arena. For instance, generating sufficiently large datasets across multiple sites and patient populations will be necessary and is one of the main obstacles to clinical implementation. This is now more feasible in today’s era whereby a vast quantity of medical imaging data are generated daily and are accessible. In fact, the fully automated methods that have started to replace manual or semi-automated methods are making analyses less labor-intensive without compromising accuracy [170,171]. Transitioning to a less labor-intensive approach will be crucial as it can be very challenging to generate the quantity of labeled data needed to both validate the approach and generalize to unseen data. Further efforts to increase the scale of investigations will, therefore, speed the era of implementing CT body composition analysis into routine clinical care.

The clinical use of AI-based body composition analysis is not only dependent on large-scale data but also on heterogenous data that are reflective of our current populations. There is a need to establish international parameters and reference ranges that guide body composition analyses in order to produce generalizable solutions for both research and clinical use. Efforts to include population reference curves that are adjusted for several demographic variables have already begun to be implemented and have been shown to have a great correlation and equivalency to manual methods [68]. Establishing international parameters is a key step in AI’s large-scale use. As a community, we will need to use demographic-conscious adjustments to allow these methods to become effective and generalizable.

The use of fully automated AI-based CT body composition analysis has great potential to revolutionize the future of medical care both within and outside of radiology. We envision that AI-based CT body composition analysis will play a crucial role in future treatment algorithms as they are widely implemented in every CT scan that is performed. Furthermore, the fully automated methods reduce the burden of analyzing clinical scans for incidentalomas and creates adequate risk assessment and prognostication that can better inform patient care. With the wider use of these automated systems, we will be able to generate more data and better be able to guide decision making, treatment planning, preoperative optimization, risk mitigation and ultimately improve patient outcomes by personalizing care without additional exposure to radiation.

Despite the great promise that this new technique presents, the vast clinical potential of CT-based body composition analysis still faces many challenges prior to widespread implementation [172,173]. For instance, there are legal liability issues with adopting fully automated AI systems. Who will be responsible for any errors that may harm patients? How can we ensure the privacy of the data? The commercialization of AI-based systems also raises ethical and equity concerns that have been covered in detail by a joint European and North American multi-society statement [174].

There are multiple ways to address these issues. The implementation of AI-based body composition analysis can begin with common problems that have ample clinical data. Multi-center consortia such as the Opportunistic Screening Consortium in Abdominal Radiology (OSCAR) can be created to clinically implement automated systems. Furthermore, studies assessing the validity of these fully automated systems and adjusting for various demographic features will aid in their widespread and equitable implementation [172]. Establishing body composition parameters and reference ranges based on age, sex and race is of utmost importance.

It is also worth noting that there have been other new measures used to assess abdominal obesity. For instance, waist-to-height ratio (WHtR) has been shown to be a better screening tool than BMI and waist circumference for predicting cardiometabolic risk [175,176]. Other measures such as body shape index (ABSI), conicity index (CI) and body roundness index (BRI) are newer central obesity indexes, and data show that some of those measures are better suited for specific populations [177,178,179,180]. Other novel measures include the lipid accumulation product (LAP), which uses triglyceride and waist circumference; and the triglyceride-glucose index (TyG index), which utilizes fasting blood glucose and fasting triglyceride [181]. Despite the usefulness of those measures, the literature is sparse on how they fare with CT-based body composition techniques. Given that these methods are cheap, future studies showing how they compare to CT-based body composition may be warranted, especially for predicting obesity-related metabolic disorders. Moreover, the use of these clinical data in conjunction with CT-based AI models will be an interesting direction, since their incorporation may aid in strengthening the analysis of body composition, which aims to better prognosticate patients for better clinical decision making and improved patients’ outcomes.

## 6. Conclusions

Advances in deep learning have led to excellent speed and accuracy in analyzing body composition on CT scan. Several fully automated CT-based body composition analyses have been developed and have shown great promise toward potential widespread investigations and clinical implementation. Multiple studies have shown the great potential of CT body composition analysis, especially in identifying patients at risk of complications. This can potentially improve the current risk prediction models and contribute to better clinical outcomes. CT body composition analysis can potentially help us personalize and tailor therapy by selecting safer alternative approaches to decrease complications and mitigate risk. Further studies are still needed to validate existing models and ensure their generalizability prior to widespread clinical use. Nonetheless, this review provides the current state of the art applications of CT body composition and suggests future directions and considerations to guide novel investigations and widespread clinical implementation.

## Figures and Tables

**Figure 1 diagnostics-13-00968-f001:**
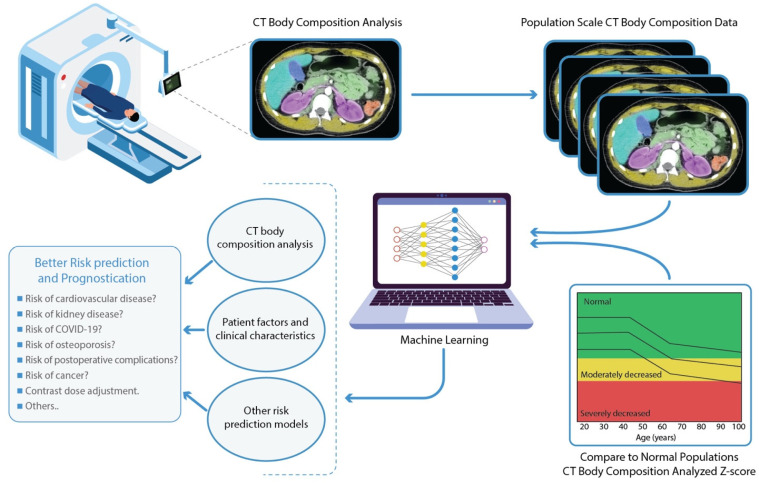
Population-scale machine learning-based CT body composition analysis for better risk prediction and prognostication. Body composition metrics can be extracted automatically from abdominal CT scans using machine learning-based segmentation approaches. Population scale CT body composition analysis can help establish age-, sex-, and race-specific Z-scores and reference curves for each metric. Patient-specific CT body composition metrics can be adjusted based on the reference curves prior to incorporation into risk prediction or prognostication models to aid in improved clinical decision-making.

**Table 1 diagnostics-13-00968-t001:** Summary of the standard metrics analyzed from CT body composition: Several metrics can be obtained from CT body composition analysis using unique ways of technical calculations. Slice identification is commonly done with the support of DenseNet or ResNeXt which is a multi-class natural image classification architecture that can help find the optimal slice. CT body composition analysis is then performed using a tissue segmentation model, commonly based on the U-Net model which is highly effective for medical image segmentation. Each metric has an important value in the clinical arena guiding risk prediction and prognostication with the potential to optimize patients outcomes.

CT Body Composition Metrics	Analysis Method	Terminology of an Abnormal Value	Clinical Applications
Skeletal Muscle Index (SMI) (in cm^2^/m^2^)	Localization and Segmentation of Skeletal muscle at the appropriate location (commonly L3) followed by calculation of the total skeletal muscle cross-sectional area divided by height squared, resulting in SMI calculation	Sarcopenia	Predict postoperative outcomes and the risk of various disease outcomes including cancer, cirrhosis, Inflammatory bowel disease, kidney disease, Severe COVID-19 and critical illness [16,17,18,19,20,21,22,23,24,25,26,27].
Skeletal Muscle Density (in HU)	After muscle segmentation, calculation of the mean muscle radiation attenuation of a muscle tissue excluding inter- and intra- muscular adipose tissue. This gives a muscle density expressed in Hounsfield units (HU). A higher attenuation indicates a low muscle density.	Myosteatosis or low muscle quality or muscle fat infiltration	Associated with poor metabolic function and worse perioperative morbidity and mortality. Can predict the risk of long-term oncological outcomes specially in those receiving treatments. It’s also an independent predictor of mortality in necrotizing pancreatitis, COVID-19 and those undergoing hemodialysis [28,29,30,31,32,33,34].
Adipose Tissue - Subcutaneous (SAT) - Visceral (VAT)(in cm)	CT slice from an appropriate location is segmented and a region of interest(ROI) pass through the abdomen separating the abdominal wall from fat in a smooth manner due to the high difference in density and intensity, thus separating SAT from VAT. Automated analysis of a ROI that includes all similar grey pixels of VAT then results in a sizable area.	- Visceral adiposity - Subcutaneous adiposity - Sarcopenic Obesity - Ratio of Visceral-to- subcutaneous fat (V/S) (cm^3^/cm)	Predictor of major cardiovascular events, nonalcoholic fatty liver cirrhosis, kidney disease, cancer, metabolic syndrome, severe COVID-19 and mortality in asymptomatic screening population [28,29,35,36,37,38]
Bone Mineral Density (BMD) (in HU)	The mean vertebral BMD is measured by placing a ROI commonly in L1-L3 vertebral bodies at the coronal, sagittal and axial images. Automated analysis of the cortical and trabecular area/BMD is obtained in HU.	- Osteopenia - Osteoporosis	Can accurately screen for osteoporosis and predict future risk of osteoporotic fractures. Can also aid with measurement of syndesmophytes and predict progression of ankylosing spondylitis [39,40,41,42,43]

## Data Availability

Not applicable.

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
