# Peer review of "Role of Machine Learning-Based CT Body Composition in Risk Prediction and Prognostication: Current State and Future Directions"

_diagnostics, 2023, doi:10.3390/diagnostics13050968_

Round 1

Reviewer 1 Report

(1) The submission hasn't new insight and no cotribution in the associated domain;

(2) Authors should use figures and tables to express their thoughts.

Author Response

POINT 1: The submission hasn't new insight and no contribution in the associated domain.

RESPONSE 1: We thank the reviewer for taking the time to read our work and regret that they did not find it to have new insights or contributions. We have tried our best to focus on the present state of the art and highlight the advances of artificial intelligence incorporation in extending the versatility of CT imaging. Furthermore, we also offer future directions to guide the field forward. We hope that after incorporating the other two reviewers’ suggestions and table you find our work more insightful.

POINT 2: Authors should use figures and tables to express their thoughts.

RESPONSE 2: We appreciate the reviewer comment to use tables and figures to express our thoughts. We have added a table that summarize the standard metrics obtained from CT body composition and hope it helps the readers quickly understand the main content. Additionally we have a figure that summarize the idea of the review article and explain the importance of population-scale Machine learning-based CT body composition analysis for better Risk prediction and prognostication  of various clinical outcomes.

Reviewer 2 Report

Manuscript Title: Role of machine learning-based CT body composition in risk prediction and prognostication: current state and future directions

This manuscript reviews the application of machine learning in CT body composition. It starts by reviewing the existing metrics that could be retrieved from CT body composition, then introduces some applications in the clinic environment, and finally outlines the possible future research direction for applying machine learning in CT-based body composition. The reviewer thinks this paper could be accepted for publishing in the journal (if the journal is willing to embrace the literature review-type paper), provided that the author has addressed the following comments.

(1) Section 2: the author is encouraged to use a table or plot to visualize the standard metrics reviewed in this section. In the meantime, the author could add associated references and comments in the table/plot. Such a visualization would help the reader quickly understand the main content.

(2) Sections 3 and 4: the reviewer fails to see the connection between these two sections. Could the author add more transitions to link these two sections smoothly?

(3) Section 5: the author is highly encouraged to review and discuss the gap between the current clinic situation and the possible application of machine learning. Meanwhile, please discuss the obstacles/difficulties (e.g., hard to obtain the data, etc.) that prevent applying machine learning.

(4) Section 6: the author should explicitly outline the importance of this review and how this review could benefit the research community.

Author Response

Point 1: Section 2: the author is encouraged to use a table or plot to visualize the standard metrics reviewed in this section. In the meantime, the author could add associated references and comments in the table/plot. Such a visualization would help the reader quickly understand the main content.

RESPONSE: We appreciate the reviewer suggestion to incorporate a table that summarize the standard metrics obtained from CT body composition. We have added the table to the review article and hope it helps the readers quickly understand the main content.

POINT 2: Sections 3 and 4: the reviewer fails to see the connection between these two sections. Could the author add more transitions to link these two sections smoothly?

RESPONSE: We appreciate the reviewer’s helpful suggestion to improve the readability and have added a transition between the two sections.

POINT 3: Section 5: the author is highly encouraged to review and discuss the gap between the current clinic situation and the possible application of machine learning. Meanwhile, please discuss the obstacles/difficulties (e.g., hard to obtain the data, etc.) that prevent applying machine learning.

RESPONSE: We appreciate the reviewer for appreciating our attempt to review and discuss the research gaps. We have expanded on some of the obstacles/difficulties that were introduced in the prior versions.

POINT 4: Section 6: the author should explicitly outline the importance of this review and how this review could benefit the research community.

RESPONSE: We thank the reviewer for their helpful suggestion. We have added a sentence in the end of the conclusion clarifying the importance of this review and how it can help the research community.

Reviewer 3 Report

Review for the manuscript entitled Role of Machine Learning-Based CT Body Composition in Risk Prediction and Prognostication: Current State and Future Directions.

This review article highlights how machine learning, which has made remarkable progress in recent years, has extended the versatility of CT imaging: body composition assessed by CT is not only associated with obesity-related metabolic disorders and sarcopenia, but also reflects the risk of atherosclerotic disease, osteoporosis and inflammatory bowel disease. AI-enhanced diagnostic imaging is a topic of recent interest in the medical community, making this review valuable.

On the other hand, the reviewer requests several additions in this manuscript.

1- What are the specific criteria for determining sarcopenia on CT? Absolute muscle mass, muscle mass corrected for height squared, ratio to fat and bone, etc. are envisaged, but what are the most effective candidate?

2- The usefulness of some measures for assessing abdominal obesity has been reported. Examples include waist-to-height ratio (WHtR), A body shape index (ABSI), conicity index and body roundness index. Compared to these indices, which cost nothing at all, is CT better at detecting the risk of obesity-related metabolic disorders? If there is evidence, please add it. Alternatively, if not, please present it as an issue.

3- To promote CT, why not also mention the decreasing radiation exposure per unit volume of imaging?

4- Apart from body composition, can CT imaging of the brain predict the onset of dementia?

Thats all.

Author Response

POINT 1: This review article highlights how machine learning, which has made remarkable progress in recent years, has extended the versatility of CT imaging: body composition assessed by CT is not only associated with obesity-related metabolic disorders and sarcopenia, but also reflects the risk of atherosclerotic disease, osteoporosis and inflammatory bowel disease. AI-enhanced diagnostic imaging is a topic of recent interest in the medical community, making this review valuable.

On the other hand, the reviewer requests several additions in this manuscript.

1- What are the specific criteria for determining sarcopenia on CT? Absolute muscle mass, muscle mass corrected for height squared, ratio to fat and bone, etc. are envisaged, but what are the most effective candidate?

RESPONSE: We thank the reviewer for their positive comments and appreciate their helpful questions. We have included a paragraph of the specific criteria used for determining sarcopenia on CT at the end of the “Muscle Mass” section. While there is no consensus on the most effective candidate in the literature, the majority of studies appear to use skeletal muscle index at the L3 to assess sarcopenia with several published specific cut-offs.

POINT 2: The usefulness of some measures for assessing abdominal obesity has been reported. Examples include waist-to-height ratio (WHtR), A body shape index (ABSI), conicity index and body roundness index. Compared to these indices, which cost nothing at all, is CT better at detecting the risk of obesity-related metabolic disorders? If there is evidence, please add it. Alternatively, if not, please present it as an issue.

RESPONSE: We thank the reviewer for this very helpful suggestion. We have included discussion on these newer measures in the end of section 5, as suggested. 

 POINT 3: To promote CT, why not also mention the decreasing radiation exposure per unit volume of imaging?

RESPONSE: We appreciate the reviewer’s helpful suggestion to promote CT use. We have added a sentence in the end of the introduction incorporating this suggestion.

POINT 3: Apart from body composition, can CT imaging of the brain predict the onset of dementia?

RESPONSE: We thank the reviewer for this insightful question. The literature is still sparse with regards to AI-based methods for detecting the onset of dementia, and most studies are focusing on clinical data or MRI (https://www.ncbi.nlm.nih.gov/pmc/articles/PMC9405227/), which is more sensitive than CT for dementia. We believe that this is outside the scope for our manuscript, especially with the very limited data and our targeted focus on body composition.